# Multiple Drug Resistance Patterns in Various Phylogenetic Groups of Hospital-Acquired Uropathogenic *E. coli* Isolated from Cancer Patients

**DOI:** 10.3390/antibiotics9030108

**Published:** 2020-03-02

**Authors:** Ahmed Talaat Mahmoud, Mohamed Taha Salim, Reham Ali Ibrahem, Adel Gabr, Hamada Mohamed Halby

**Affiliations:** 1Department of Microbiology and Immunology Faculty of Pharmacy Al-Azhar University, Assuit 71524, Egypt; ahmedabdel-al@azhar.edu.eg (A.T.M.); mohamedsalem@azhar.edu.eg (M.T.S.); 2Department of Microbiology and Immunology Faculty of Pharmacy El-Minia University, Minia 61111, Egypt; Reham.ali@mu.edu.eg; 3Department of Medical Oncology and Malignant Hematology, South Egypt Cancer Institute, Assuit University, Assuit 71515, Egypt; Adelgabre@yahoo.com

**Keywords:** uropathogenic *E. coli*, phylogenetic analysis, drug resistance

## Abstract

Cancer patients are more susceptible to several bacterial infections, particularly urinary tract infections caused by uropathogenic *E. coli* (UPEC). The objective of this work was detection and the phylogenetic characterization of hospital-acquired isolates of uropathogenic *E. coli* in cancer patients and the determination of its relation with antibiotic resistance. A total of 110 uropathogenic *E. coli* responsible for hospital-acquired urinary tract infections in cancer patients were included in this study. A triplex PCR was employed to segregate different isolates into four different phylogenetic groups (A, B1, B2 and D). Drug resistance was evaluated by the disc diffusion method. All of the isolates were multiple drug-resistant (MDR) and 38.18% of all UPEC isolates were extended-spectrum beta-lactamase (ESBL) producers from which 52% were positive for the *bla*CTX-M gene, 40% for the *bla*TEM gene, and 17% for the *bla*SHVgene. Among 42 ESBL-producing uropathogenic *E. coli* isolates, the majority belonged to phylogenetic group B2 (43%), followed by group D (36%), group A (19%) and group B1 (2%). Our results have shown the emergence of MDR isolates among uropathogenic *E. coli* with the dominance of phylogenetic group B2. Groups A and B1 were relatively less common. The most effective drug in all phylogenetic groups was imipenem.

## 1. Introduction

*Escherichia coli* (*E. coli*) is the most prevalent commensal inhabitant of the gastrointestinal tract (GIT) of humans and animals. It is a common pathogen linked with community-associated as well as hospital-acquired infections [1]. Hospital acquired infections (HAIs) are the major cause of life-threatening complications in hospitalized patients, especially immunocompromised patients, such as cancer patients [2]. Urinary tract infections (UTI) caused by *E. coli* strains are the most common cause of morbidity in hospitalized cancer patients due to their several impairments of host defense [3].

Cancer is the leading cause of death worldwide. The most important risk factors for HAIs in cancer patients are surgical operations, extensive use of medical devices and immunosuppression [4].

The cancer patient is immunocompromised due to the underlying malignancy, such as leukemia, and also due to the destructive complications of cancer treatment, such as chemotherapy, radiation, and bone marrow transplantation. This could lead to prolonged immunosuppression, increasing the risk of infection and possibly worsening the prognosis [5].

The increased resistance of *E. coli* to antibiotics has been frequently reported from different regions of the world as a complication of treatment. Antibiotics given empirically without suitable antibiotic sensitivity testing are considered as the major causes for the emergence of multidrug resistance. The current awareness of the organism that causes UTI and their antibiotic susceptibility is obligatory to ensure appropriate therapy. The dissemination of ESBL-producing *E. coli* in the hospital setting is a problem with major therapeutic and epidemiologic consequences [6].

*E. coli* was classified into four major phylogenetic groups: A, B1, B2 and D. The phylogenetic analysis was done by a simpler method targeting three genetic markers: chuA (required for heme transport), yjaA (unknown function) genes and a DNA fragment, TSPE4.C2. The virulent strains, including UPEC, belong to phylogenetic groups B2 and D and the less virulent and commensal isolates belong to B1 and A [7].

The evolution and spread of various phylogenetic groups of antibiotic-resistant *E. coli* became a worldwide health concern in human medication, so the assessment of the phylogenetic distribution of antimicrobial-resistant *E. coli* is vital for therapeutic and economic functions [8]. Thus, according to the importance of different *E. coli* phylogenetic groups and the role of its antibiotic resistance pattern. The aim of this study is the detection of phylogenetic groups of uropathogenic *E. coli*, determination of the antibiotic resistance profile of uropathogenic *E. coli*, and assessment of the relation between phylogenetic groups and antibiotic resistance of uropathogenic *E. coli.*

## 2. Results

The total number of urine samples received during the study period (from March 2018 to March 2019) was 212, out of which significant bacterial growth was observed in 152 (71.7%) samples. Out of 152 urine samples, *E. coli* was isolated from 110 (72.4%) samples (Figure 1).

### 2.1. Antimicrobial Sensitivity Pattern of Uropathogenic E. coli Isolates

The antibacterial susceptibility profile was examined in all 110 uropathogenic *E. coli* isolates. The resistance rate to each antibiotic was calculated as the number of resistant isolates divided by the total number of isolates (Table 1).

### 2.2. Phenotypic Detection of ESBL

A total of 51 *E. coli* isolates (46.63%) were found as potential ESBL producers, they showed reduced susceptibility to one or more of ceftazidime, aztreonam, cefotaxime or ceftriaxone. Out of 51 isolates that were considered as potential ESBL producers, 42 (38.18%) were confirmed as ESBL producers.

### 2.3. Genotypic Detection of ESBL Producers by Polymerase Chain Reaction

All confirmatory screened uropathogenic *E. coli* isolates were analyzed by PCR for the detection of ESBL genes. It was found that CTX-M was the main type of ESBL, TEM was the second, and then SHV as shown in Table 2 and some strains had more than one genotype of ESBL genes. The detection of these genes by gel electrophoresis is shown in Figure 2.

### 2.4. Distribution of Phylogenetic Groups in ESBL Producer-UPEC Isolates

Strains were assigned to phylogenetic groups based on the presence or absence of the three genes: *chuA*, YjaA, and TspE4.C2. Phylogenetic analysis of isolates indicated that the majority of uropathogenic *E. coli* isolated from suspected cases of UTI of cancer patients belonged to group B2 and D, as shown in Table 3. The detection of these genes by gel electrophoresis is shown in Figure 3.

## 3. Discussion

Cancer patients are known to be susceptible to various nosocomial infections due to the destructive complications of cancer treatment on their immune system [4]. Urinary tract infection (UTI) is one of the major causes of morbidity in cancer patients. *E. coli* was the most common organism isolated in cancer patients with UTI [3].

In this study, the urine culture was taken from cancer patients, which showed that 71.7% were positive. This result is similar to that reported by Tancheva et al. [9] in Varna where the rate was 68%. However, it is higher than that reported by Yakovlev et al. [10] in India, where the rate was 33.4% and Raad et al. [11] in Finland found that UTI was present in 12.5% of cancer patients.

This study revealed that the main isolated organisms from urine culture taken from cancer patients were *Escherichia coli* (72.4%). This is comparable with previous studies reported by Bhusa et al. [12] in the USA, Tancheva et al. [9] in Varna, Mukta et al. [13] in India, and Chandra et al. [14] in India but with lower rates than our study, where the rates were 69.5%, 64.7%, 38.09%, 37.5%, respectively.

Antibiotic resistance is a major clinical problem when treating UTI in cancer patients caused by UPEC. The resistance to imipenem seen in our study, which is more than any other previous study, perhaps finds a logical explanation due to the frequent use of imipenem as routine treatment for resistant strains. Other studies reported lower rates of resistance to imipenem, such as Sedighi et al. [15] in Iran, who reported it to be 3.3% and Mukta et al. [13] in Bulgaria, who found the resistance to be 9%. In addition, Elsayed et al. [16], in Egypt, reported that the resistance rate was 2%. The lower resistant rates for imipenem are probablybecause it is a very powerful drug used only in hospital settings and not as first-line therapy in out-patients clinics [17].

In our study, *E. coli* isolates exhibited maximal resistance against ceftriaxone. This finding is quite challenging because ceftriaxone is a commonly-used empirical therapy in most hospitals. This result is similar to that reported by Mahgoub et al. [18] in Egypt, where the rate was 79.6%. However, the result in this study is higher than that reported by Abdel-Moaty et al. [19] in Egypt and Khan et al. [20] in Bangladesh, where the rates were 61% and 41.9%, respectively.

The regional variations of resistance to antibiotics may be explained by different local antibiotic practices. The influence of inappropriate antibiotic use on the event of antibiotic-resistant strains, especially broad-spectrum agents, has been proven through empirical observation [21].

Extended-spectrum β-lactamase (ESBL) production is an important resistance mechanism that inhibits the antimicrobial actions against infections caused by Enterobacteriaceae. ESBLs are considered a serious threat to the currently available antimicrobial agents [22]. The prevalence of bacteria producing ESBLs varies worldwide, with reports from North America, Europe, South America, Africa and Asia [23].

Preliminary detection of ESBL-producing *E. coli* isolates was done by screening tests according to CLSI (2016) that depend on reduced susceptibility to one or more of cefotaxime, ceftazidime, aztreonam, cefotaxime or ceftriaxone. Accordingly, 46.63% (51/110) of UPEC isolates were considered as potential ESBL producers. This is comparable with Shakya et al. [24] in Nepal who reported that 43.8% of UPEC isolates were potential ESBL producers by screening tests.

In addition, a similar result was obtained by Alqasim et al. [25] in Saudi Arabia who reported that 41% of UPEC isolates were potential ESBL producers, but a higher rate of potential ESBL-producing UPEC was reported by Al-Mayahie et al. [26] in Iraq, Thabit et al. [27] in Egypt and Mukherjee et al. [28] in India, where the rates were 80.2%, 76.47, 70%, respectively. For the ESBL confirmatory double-disk synergy test, DDST detected ESBLs in 42/110 (38.18%). This percentage is similar to the result by Alqasim et al. [25] in Saudi Arabia who reported that 33.3% of UPEC isolates were confirmed ESBL producers by DDST. Furthermore, Islam et al. [29] in Bangladesh reported a comparable result where 32% of UPEC isolates were confirmed ESBL producers by the same test.

Higher levels of ESBL production were reported by Al-Mayahie et al. [26] in Iraq, Chandra et al. [14] in India, Al-Agamy et al. [30] in Egypt, Mekki et al. [31] in Sudan, and Abayneh et al. [32] in Southwest Ethiopia. The rates reported for each study were 64.8%, 62.5%, 60.9%, 53% and 76.5%, respectively. In contrast, lower results were mentioned by Sedighi et al. [15] in Iran (27.3%) and Villanueva et al. [33] in the Philippines (12.4%).

In our study, a higher degree of resistance was shown by ESBL producers than ESBL non-producers. The obtained results revealed that the resistance level to all cephalosporins (cefotaxime, ceftazidime, ceftriaxone, and cefepime) and aztreonam was significantly higher in ESBL-producing *E. coli* in comparison with non-ESBL-producing isolates (*p* < 0.001). This finding is in accordance with other reports, such as Islam et al. [29] in Bangladesh, and Abdel-Moaty et al. [19] in Egypt.

In the current study, ESBL-producing isolates exhibited significantly higher resistant rates to non-β-lactamase antimicrobials agents including fluoroquinolone, aminoglycosides, tetracycline and trimethoprim/sulfamethoxazole, compared to non-ESBL-producing isolates. The possible explanation for this observation may be that ESBLs are encoded on plasmids and can be mobile and therefore, easily transmissible as resistance gene elements for other antimicrobials from one organism to another.

In the present study, the genotyping of ESBL-producing UPEC isolates was done by PCR to determine the most common ESBL genes responsible for resistance. We reported that CTX-M was the main ESBL type (52%), followed by TEM (40%), then SHV (17%). Twelve isolates had more than one type of ESBL, where CTX-M + TEM were found in 7 (16.67%) isolates, CTX-M + SHV were observed in 11.9% of isolates and 4.76% of isolates had TEM + SHV genes. The same order of gene type presence, but with different percentages, was reported by Chakraborty et al. [34] in India, where the CTX-M gene was detected in 88% of *E. coli*, followed by TEM (19%) and SHV (2%). In addition, the same pattern was mentioned by Zhao et al. [35] in China, who reported that the rate of CTX-M, TEM, and SHV among *E. coli* isolates was 42.5%, 4.2%, and 0.8%, respectively.

The increase of consumption of cefotaxime and ceftazidime could have contributed to the emergence of CTX-M enzymes encoding genes among *E. coli* strains in Egyptian hospitals [36].

CTX-M β-lactamases constitute a novel and rapidly growing family of plasmid-mediated ESBLs, which are currently replacing mutant TEM or SHV ESBL families [6].

In contrast, Azargun et al. [37] in Iran, reported that the TEM gene was the major ESBL gene in UPEC isolates, followed by the CTX-M gene and the SHV gene, where the rates for TEM, CTX-M, and SHV were 75.6%, 78.6% and 33.3%, respectively.

In our study, Triplex PCR-based phylogenetic analysis was carried out for EP-UPEC isolates according to the method described by Clermont et al. [7]. Phylogenetic grouping revealed that most of the isolates belonged to the B2 group (*n* = 18, 43%), followed by group D (*n* = 15, 36%), group A (*n*= 8, 19%) and group B1 (2%). The majority of studies concerning the phylogenetic grouping among UPEC have reported a similar distribution, such as Zhao et al. [35] in china, Abdi et al. [38] in Iran, Lee et al. [39] in Korea, Johnson, and Stell [40] in Minnesota, Picard et al. [41] in France, Ejrnæs et al. [42] in Denmark, Kanamaru et al. [43] in Japan, and Alghoribi et al. [44] in England.

However, a few studies have reported a different distribution of phylogenetic groupings for UPEC isolates, such as Marialouis et al. [45] in India who found that most of the isolates belonged to B2, followed by A, B1, and D, and phylogenetic group D isolates were the least frequent. In addition, Bashir et al. [46] in Pakistan observed the same finding with their phylogenetic analysis.

In contrast, some studies observed that most of the UPEC isolates belonged to phylogenetic group D, followed by A, B1, and B2, such as Adwan et al. [47] in Palestine and Abdallah et al. [48] in china, where phylogenetic group B2 isolates were the least frequent phylogenetic group.

The results of drug resistance according to phylogenetic groups are shown in Table 4. Our findings showed that group B2 was the most predominant phylogenetic group and most resistant strain to commonly used antibiotics among patients. This finding is in agreement with other studies such as Iranpour et al. [49] in Iran. However, Bashir et al. [46] in Pakistan reported that group B2 isolates exhibited lower levels of drug resistance than our study.

In our study, group D isolates were totally resistant to ceftriaxone, aztreonam, and ceftazidime, but highly resistant to cefotaxime (93.3%), cefepime (86.8%), sulfamethoxazole/trimethoprim (73.33%) and ciprofloxacin (66.67%), and less resistant to gentamicin (46.4%), imipenem (33.3%), amikacin (20%), nitrofurantoin (20%) and levofloxacin (40%). This result is similar to Bashir et al. [46]. In contrast, Iranpour et al. [49] in Iran found that group D isolates exhibited a low level of drug resistance.

In this study, Group A isolates were totally sensitive to imipenem and sulfamethoxazole/trimethoprim, less resistant to gentamicin (25.4%), amikacin (12.5%), nitrofurantoin (37.5%), levofloxacin (37.5%), colistin (12.5%) and ciprofloxacin (12.5%), and highly resistant to ceftriaxone (87.5%), ceftazidime (75%), cefotaxime (87.5%), cefepime (62.5%) and aztreonam (50%). In contrast, Bashir et al. [46] in Pakistan found that group A isolates showed a higher level of drug resistance than our study.

In our study, we observed a significant difference between phylogenetic groups and resistance to different groups of antibiotics. This finding is similar to the study reported in china by Wang et al. [50]. In contrast, Cristea et al. [51] in Romania found that their statistical analyses did not reveal any statistical significance of the correlation between antibiotic resistance and *E. coli* phylogenetic groups.

The frequency *of blaCTX-M*, *blaTEM* and *bla SHV* ESBL genes in phylogenetic groups is summarized in Table 5. Our phylogenetic analysis of the isolates revealed that strains harboring CTX-M gene were associated with the D phylogenetic group and there is a significant difference was observed in the frequency of CTX-M type between the phylogenetic groups. 

## 4. Materials and Methods

### 4.1. Collection of Samples

Urine samples were obtained from cancer patients suffering from urinary tract infection admitted at Assuit university hospitals from March 2018 to March 2019. Mid-stream urine samples were collected in sterile, dry containers after cleaning the genital area [52]. The processing of collected urine samples was done quickly to avoid contamination. Samples that could not be processed immediately were refrigerated at 4 °C for a few hours [53].

### 4.2. Microscopic Examination of Urine Samples

The collected urine samples were examined microscopically for pus cell count by high power field (HPF); pyuria means >5–10 leukocytes/HPF [54]. A gram-stained smear of the uncentrifuged urine samples was examined. A positive smear was defined by the presence of more than two bacteria per oil immersion field [55].

### 4.3. Viable Count

Viable Count was performed for urine samples using the pour plate method. The presence of ≥10^5^ CFU/mL was considered as significant bacteriuria, whereas lower numbers of organisms were considered as insignificant bacteriuria [56].

### 4.4. Isolation and Identification of E. coli Isolates

Urine samples were centrifuged at 4000 rpm for 5 min. The sediment was streaked on CLED agar medium, EMB medium and MacConkey agar medium (Oxoid, Basingstoke, UK). Then these media were incubated at 37 °C for 24 h. The isolated bacteria were then identified by using Gram stain and their biochemical characteristics. These included an indole production test, methyl-red test, Voges–Proskauer test, citrate utilization test, triple sugar iron agar test, and sugar fermentation patterns (Oxoid, Basingstoke, UK) [57].

### 4.5. Antimicrobial Susceptibility Patterns of Uropathogenic E. coli Isolates

All isolates were screened for susceptibility to twelve antimicrobial agents, namely amikacin (AK, 30 µg), gentamicin (CN,10 µg), aztreonam (AZT, 30 µg), cefepime (FEP, 30 µg), cefotaxime (CXT, 30 µg), ceftazidime (CAZ, 30 µg), ceftriaxone (CRO, 30 µg), imipenem (IMP, 10 µg), ciprofloxacin (CIP, 5 µg), levofloxacin (LEV, 5 µg), nitrofurantoin (F, 300 µg), trimethoprim/sulfamethoxazole (SXT, 1.25/23.75 µg) and colistin sulfate powder. All discs were supplied from (Bioanalyse, Ankara, Turkey)while colistin sulfate powder was supplied from (Merck, Darmstadt, Germany) All antimicrobial agents were determined by the disc diffusion method (except colistin sulfate) in accordance with the guidelines of the Clinical and Laboratory Standards Institute [58]. The results was interpreted according to Clinical Laboratory Standard Institute [59]. Broth microdilution was performed to determine the minimum inhibitory concentration (MIC) of colistin in cation-adjusted Mueller Hinton broth according to Clinical and Laboratory Standards Institute (CLSI) guidelines [60]. European Committee on Antimicrobial Susceptibility Testing (EUCAST) breakpoints were used for interpretation of colistin MIC results (with a susceptible breakpoint of al Susceptibility Testing (EUCAST) breakpoint >2 mg/L) [61].

### 4.6. Phenotypic Detection of Extended-Spectrum β-Lactamases

#### 4.6.1. Screening of ESBL-Production

UPEC isolates that displayed decreased susceptibility to one or more of ceftazidime, aztreonam, cefotaxime or ceftriaxone were considered as potential ESBL-producing isolates according to CLSI, 2016 [59].

#### 4.6.2. Double-Disc Synergy Test (DDST)

A Muller–Hinton agar plate (Oxoid, Basingstoke, UK) was inoculated with bacterial suspension matched with 0.5 McFarland turbidity standards to nearly 10^8^ CFU/mL as recommended for the standard disc diffusion susceptibility test. The following discs, namely ceftazidime (30 μg), aztreonam (30 µg), cefotaxime (30 μg), and ceftriaxone (30 μg), were placed 20 mm (center to center) from the amoxicillin/clavulanic acid disc (20 μg/10 μg). Following incubation for 24 h at 37 °C, the enhancement of the zone of inhibition between a β-lactam disc and that containing the β-lactamase inhibitor was indicative for the presence of an ESBL [62].

### 4.7. Genotypic Detection of ESBL Genes by a Polymerase Chain Reaction

#### 4.7.1. Extraction of Genomic DNA from Bacterial Culture

The Quick DNA universal kit was used for Genomic DNA extraction according to the protocol provided by the manufacturer (Zymo Research, Irvine, CA, USA, catalog No. D 4068).

#### 4.7.2. Amplification of DNA by PCR

The primer sequences used for the detection of *bla TEM*, *bla SHV* and *bla CTXM* are shown in Table 6. The PCR reaction for each gene was performed in a final volume of 25 µL containing 12.5 µL of 2x master mix (BIOLINE, London, UK), 1 µL of DNA template, 9.5 µL of sterile distilled water and 1 µL of each primer in a thermal cycler (Biometra, UNO-Thermo block, Macclesfield, UK).

The following conditions were used for amplification of the *bla CTXM* gene: an initial denaturation step at 94 °C for 2 min, followed by thirty-five cycles consisting of denaturation at 95 °C for 20 s, annealing at 53 °C for 30 s and extension at 72 °C for 30 s; and with a final extension at 72 °C for 3 min. The conditions for amplification of the *bla TEM* gene were as follows: an initial denaturation at 96 °C for 5 min, followed by thirty-five cycles of 96 °C for 1 min, 44 °C for 1 min, and 72 °C for 1 min; and with a final extension at 72 °C for 10 min. While the conditions for amplification of the *bla SHV* gene were as follows: an initial denaturation at 96 °C for 5 min, followed by thirty-five cycles of 96 °C for 1 min, 59 °C for 1 min, and 72 °C for 1 min; and with a final extension at 72 °C for 10 min.

#### 4.7.3. Phylogenetic Group Typing

Strains were assigned to one of the four *E. coli* phylogenetic groups (A, B1, B2 and D) using a triplex PCR based on the presence or absence of two marker genes (chuA and yjaA) and the DNA fragment TSPE4.C2. According to the PCR-based method described by (Clermont et al., 2000), the primer sequences used for detection of *chuA, yjaA* and the DNA fragment *TSPE4.C2* are shown in Table 7).

The triplex PCR assays for phylogenetic group typing were performed in a final volume of 25 µL containing 12.5 µL of 2x master mix (BIOLINE, London, UK), 2 µL of DNA template, 8.5 µL of sterile distilled water and 1 µL of each primer in a thermal cycler (Biometra, UNO-Thermo block). DNA amplification was carried out according to the following thermal conditions: Initial denaturation at 94 °C for 4 min, thirty cycles consisting of denaturation at 94 °C for 5 s, annealing at 59 °C for 10 s and extension at 72 °C for 30 s, and then the final extension step at 72 °C for 5 min.

#### 4.7.4. Detection of PCR Products by Agarose Gel Electrophoresis

The PCR products were separated by 2% agarose gel electrophoresis (ApplichemGmbh, Darmstadt, Germany), stained with ethidium bromide (Sigma-Aldrich, Germany) for 45 min under 80 V in 1X trisborate EDTA (TBE) buffer and visualized by ultraviolet Tran’s illuminator (HeroLab UVT-20M, Wiesloch, Germany).

### 4.8. Statistical Analysis

The collected data were statistically analyzed using the Statistics Package for Social Sciences (SPSS) version 21. The difference was considered to be statistically significant when *p* ≤ 0.05.

## 5. Conclusions

EP-UPEC strains showed multidrug resistance and the most effective drug was imipenem. CTX-M was the most prevalent ESBL genotype and the majority of EP-UPEC strains had more than one ESBL gene.

Our findings showed that group B2 and group D were the most predominant phylogenetic groups among cancer patients infected with UPEC. In addition, we observed that certain polygenetic groups are more resistant than others, which could be due to greater exposure of certain phylogenetic groups to antimicrobials. Other studies among cancer patients in other regions are needed to provide a greater understanding of the prevalence of antimicrobial drug resistance and the geographic distribution of *E. coli* phylogenetic groups.

Regular study of antibiotic resistance patterns among cancer patients will help clinicians to prescribe the most appropriate antibiotic and to avoid further development of antimicrobial drug resistance.

## Figures and Tables

**Figure 1 antibiotics-09-00108-f001:**
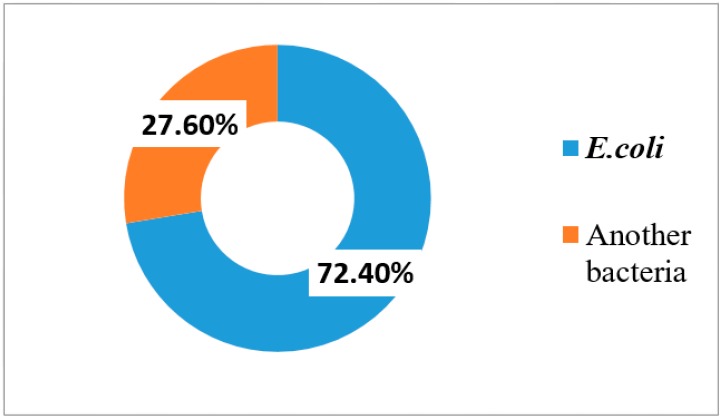
Percentage of patients with *E. coli* infection and non-*E. coli* infection among the isolates.

**Figure 2 antibiotics-09-00108-f002:**
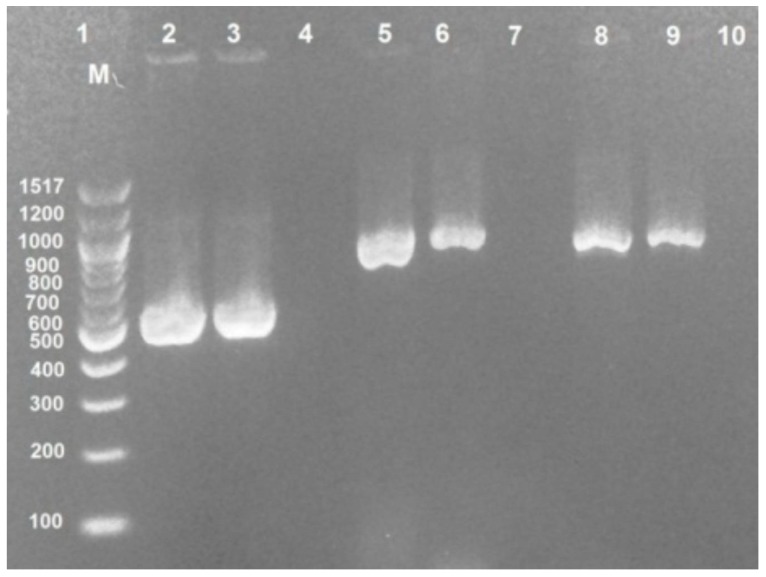
Distribution of ESBL genes in uropathogenic *Escherichia coli* isolates. Lane (1): DNA marker 100 bp. Lane (3): CTX-M gene (544 bp). Lane (6): TEM gene (867 bp). Lane (9): SHV gene (867 bp). Lane (2,5,8): Positive control (*Escherichia coli* ATCC25922). Lane (4,7,10): Negative control.

**Figure 3 antibiotics-09-00108-f003:**
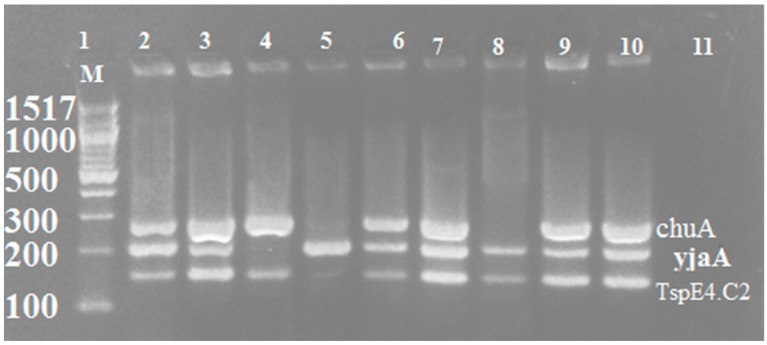
Multiplex PCR of amplified phylogenetic groups. Lane (1): DNA marker 100 bp. Lane (2, 3, 6, 7, and 9): *chuA* gene (279 bp) and *yjaA* gene (211 bp) and *TspE4*.C2 (152 bp). Lane (4): *chuA* gene (279 bp) and *TspE4*.C2 (152 bp). Lane (5): *yjaA* gene (211 bp. Lane (8): *yjaA* gene (211 bp) and *TspE4*.C2 (152 bp). Lane (10): Positive control (*Escherichia coli* ATCC25922). Lane (11): Negative control.

**Table 1 antibiotics-09-00108-t001:** Antimicrobial sensitivity pattern of uropathogenic *E. coli* isolates.

Antimicrobial Agent	Resistance Rate	Antimicrobial Agent	Resistance Rate
	No (%)		No (%)
**Ceftriaxone**	90 (81.8%)	**Amikacin**	32 (29.09%)
**Cefotaxime**	86 (78.1%)	**SXT**	88 (80%)
**Aztreonam**	78 (70.9%)	**Gentamycin**	47 (42.73%)
**Ceftazidime**	82 (74.5%)	**Ciprofloxacin**	66 (60%)
**Cefepime**	78 (70.9%)	**Levofloxacin**	50 (45.45%)
**Imipenem**	24 (21.8%)	**Colistin**	36 (32.73%)
**Nitrofurantoin**	28 (25.4%)	**-**	-

**Table 2 antibiotics-09-00108-t002:** Distribution of ESBL genes in uropathogenic *Escherichia coli* isolates.

ESBL Genes	Positive	
	No.	%
CTXM	22/42	52
TEM	17/42	40
SHV	7/42	17
CTXM + TEM + SHV	1/42	2.38
CTXM + TEM	7/42	16.67
CTX-M + SHV	5/42	11.90
TEM + SHV	2/42	4.76

**Table 3 antibiotics-09-00108-t003:** Distribution of Phylogenetic groups in EP-UPEC isolates.

Phylogenetic Group	No. of Isolates	Distribution According to Gene Grouping (*n*)	chuA	YjaA	TspE4.C2
**B2**	18 (43%)	14	+	+	+
		4	+	+	−
**D**	15 (36%)	13	+	−	−
		2	+	−	+
**B1**	1 (2%)	1	−	−	+
**A**	8 (19%)	5	−	+	−
		3	−	−	−

**[+]** = Positive, [**−**] = Negative.

**Table 4 antibiotics-09-00108-t004:** Prevalence of antimicrobial resistance in phylogenetic groups of EP-UPEC isolates.

Antimicrobial Agent	B2		D		B1		A		*p*-Value
	*N* = 18	%	*N* = 15	%	*N* = 1	%	*N* = 8	%	%
Ceftriaxone	18	100	15	100	1	100	7	87.50	0.226
Cefotaxime	18	100	14	93.33	0	0	7	87.50	**0.002 ***
Aztreonam	17	94.44	15	100	1	100	4	50.00	**0.003 ***
Ceftazidime	18	100	15	100	1	100	6	75.00	**0.030 ***
Cefepime	14	77.78	13	86.67	0	0	5	62.50	0.175
Imipenem	2	11.11	5	33.33	0	0	0	0	0.158
Nitrofurantoin	13	72.22	3	20.00	0	0	3	37.50	**0.017 ***
Amikacin	5	27.78	3	20.00	0	0	1	12.50	0.777
Gentamicin	8	44.44	7	46.67	0	0	2	25.00	0.608
Levofloxacin	9	50.00	6	40.00	1	100	3	37.50	0.629
Ciprofloxacin	14	77.78	10	66.67	1	100	1	12.50	**0.012 ***
Sulfamethoxazole-trimethoprim	15	83.33	11	73.33	1	100	0	0	**0.000 ***
Colistin	5	27.8	6	40.00	1	100	1	12.50	0.244

* *p*-Value ≤ 0.05.

**Table 5 antibiotics-09-00108-t005:** Prevalence of ESBL genes in phylogenetic groups of uropathogenic *Escherichia coli* isolates.

ESBL Genes	Total	B2		D		B1		A		*p*-Value
		*N* = 18		*N* = 15		*N* = 1		*N* = 8		
		*N*	%	*N*	%	*N*	%	*N*	%	
CTXM	*N* = 22	8	44%	11	73%	0	0	1	13%	**0.030 ***
TEM	*N* = 17	7	39%	5	33%	0	0	5	63%	0.310
SHV	*N* = 7	4	22%	1	7%	0	0	2	25%	0.556

* *p*-Value ≤ 0.05.

**Table 6 antibiotics-09-00108-t006:** PCR primers for detection of the *bla TEM* gene, *bla SHV* gene *and bla CTXM* gene.

Gene	Primer Sequence (5ʹ–3ʹ)	Size (bp)	References
***bla CTXM***	F: TTTGCGATGTGCAGTACCAGTAA	544	[63]
	R: CGATATCGTTGGTGGTGCCATA		
***bla TEM***	F: ATGAGTATTCAACATTTCCG	867	[64]
	R: CTGACAGTTACCAATGCTTA		
***bla SHV***	F: GGTTATGCGTTATATTCGCC	867	[64]
	R: TTAGCGTTGCCAGTGCTC		

**Table 7 antibiotics-09-00108-t007:** PCR primers for phylogenetic typing.

Gene	Primer Sequence (5ʹ-3ʹ)	Size (bp)	References
***chuA***	F: GAC GAA CCA ACG GTC AGG AT	279	[7]
	R: TGC CGC CAG TAC CAA AGA CA		
***yjaA***	F: TGA AGT GTC AGG AGA CGC TG	211	
	R: ATG GAG AAT GCG TTC CTC AAC		
***TspE4.C2***	F GAG TAA TGT CGG GGCATT CA	152	
	R: CGC GCC AAC AAA GTA TTA CG		

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
