# Peer review of "Multiple Drug Resistance Patterns in Various Phylogenetic Groups of Hospital-Acquired Uropathogenic E. coli Isolated from Cancer Patients"

_antibiotics, 2020, doi:10.3390/antibiotics9030108_

Round 1

Reviewer 1 Report

In this manuscript, Ahmed et al. collected urine samples from cancer patients with urinary tract infection (UTI) and performed the Double-Disc Synergy test (DDST) to screen for antibiotic resistant Escherichia coli strains. In addition, PCR assays were employed to identify their phylogenetic groups and antibiotic genes. Based on these experiments, the authors suggested that UTI caused by antibiotic resistant E. coli is highly associated with cancer patients. In addition, resistance among four E. coli polygenetic groups are significantly different, such that group B2 and D were the most resistant groups to commonly used antibiotics; while group A and B1 were relatively less resistant. The experiments in this manuscript are well performed, and the results are convincing. However, few questions need to be solved:

All 212 samples are from the same hospital, therefore, how significant the conclusion is considering there are other reports suggested there is no correlation between the antibiotic resistance and coli phylogenetic groups?

Even the conclusion is correct, what are the reasons for certain polygenetic groups are more resistant than others? If experiments are hard to perform, could the authors propose some assumptions as this is a concept manuscript?

Minor changes:

Line 22 and 23, ESBL, blaCTX-M, blaTEM, and blaSHV are not defined as they appear first time in this manuscript. Line 212, the fond size is different, the number 26 should be before the period. Line 215, add a space after the number 27. Line 216, add a space after the number 28. Line 221, the period should not be superscript. Line 226, the number 32 should be before the period. Line 250, add a space after the number 28.

Author Response

Comment 1:

All 212 samples are from the same hospital, therefore, how significant the conclusion is considering there are other reports suggested there is no correlation between the antibiotic resistance and coli phylogenetic groups?

Response to Comment 1:

Thank you for this comment….

All samples were collected from different cancer patients from this hospital which is one of the largest hospitals in this area and a large number of patients from different locations come to it. These patients are from different environments and were exposed to different antibiotics regimen before and during their stay in the hospital which made variations in antibiotic resistances and phylogenetic groups. In our study, significant correlation between phylogenetic groups and resistance to different groups of antibiotics is similar to study reported by Wang et al. [1] in china and Alwash et al. [2] Iraq.

Comment 2:

Even the conclusion is correct, what are the reasons for certain polygenetic groups are more resistant than others? If experiments are hard to perform, could the authors propose some assumptions as this is a concept manuscript.

Response to Comment 2:

Thank you for this comment…

Could be due to greater exposure of certain phylogenetic groups to antimicrobials ,so certain polygenetic groups are more resistant than others.This finding is similar to study by lee et al. [3] in Korea and Iranpour et al. [4] in Iran .

Comment 3:

Minor changes:

Line 22 and 23, ESBL, blaCTX-M, blaTEM, and blaSHV are not defined as they appear first time in this manuscript. Line 212, the fond size is different, the number 26 should be before the period. Line 215, add a space after the number 27. Line 216, add a space after the number 28. Line 221, the period should not be superscript. Line 226, the number 32 should be before the period. Line 250, add a space after the number 28. 

Response to Comment 3:

Thank you for this comment. 

All Corrections was done and clearly highlighted, using the "Track

Changes" function in Microsoft Word.

Reviewer 2 Report

The article titled “Multiple drug resistance patterns in various 2 phylogenetic groups of hospital acquired 3 uropathogenic E.coli isolated from cancer patients” reports on a characterization of several bacterial clinical isolates from patients having cancer in order to identify antimicrobial resistance. A wide screening on different antibiotics has been performed revealing serious resistance patterns in the most individuated microorganism E. coli. In the discussion section, the authors perform a comprehensive analysis considering many articles previously published reporting similar analysis in different countries.

The article is well-structured and the main aims are pretty clear.

I only suggest a review of the English and the addition of few lines in the conclusion regarding perspectives in this scenario.

Author Response

#Reviewer 2:

Comment 1:

Review of the English and the addition of few lines in the conclusion regarding perspectives in this scenario.

Response to Comment 1:

Thank you for this comment…

As recommended, we edited the manuscript and revised it Few lines in the conclusion are added as the following: Our findings showed that group B2 and group D were the most predominant phylogenetic groups among cancer patients infected with UPEC. Also, we observed certain phlyogenetic groups are more resistant than others ,this could be due to greater exposure of certain phylogenetic groups to antimicrobials. Another studies among cancer patients in other regions are needed to be done to provide greater understanding of the prevalence of antimicrobial drug resistance and geographic distribution of E. coli phylogenetic groups. Regular studying of antibiotic resistance patterns among cancer patients will help clinicians to prescribe the most appropriate antibiotic and to avoid further development of antimicrobial drug resistance

References

Wang, Y.; Zhao, S.; Han, L.; Guo, X.; Chen, M.; Ni, Y.; Zhang, Y.; Cui, Z.; He, P. Drug resistance and virulence of uropathogenic Escherichia coli from Shanghai, China. The Journal of antibiotics 2014, 67, 799.

Alwash, M.S.; Al-Rafyai, H.M. Antibiotic Resistance Patterns of Diverse Escherichia coli Phylogenetic Groups Isolated from the Al-Hillah River in Babylon Province, Iraq. The Scientific World Journal 2019, 2019.

Lee, J.; Subhadra, B.; Son, Y.J.; Kim, D.; Park, H.; Kim, J.; Koo, S.; Oh, M.; Kim, H.J.; Choi, C. Phylogenetic group distributions, virulence factors and antimicrobial resistance properties of uropathogenic Escherichia coli strains isolated from patients with urinary tract infections in South Korea. Letters in applied microbiology 2016, 62, 84-90.

Iranpour, D.; Hassanpour, M.; Ansari, H.; Tajbakhsh, S.; Khamisipour, G.; Najafi, A. Phylogenetic groups of Escherichia coli strains from patients with urinary tract infection in Iran based on the new Clermont phylotyping method. BioMed research international 2015, 2015.

Revised manuscript

Author List and Affiliations:

1.We added  middle names for authors

We added institutional email addresses of authors, province ,zip code

Abstract :

1.We deleted some words, so  abstract not exceed 200 word.

The deleted words

From line (1):Due to impairment of their host defense

From line (18) to line (21 ): The resistance rates were (81.82%) for ceftriaxone, (80%)for sulfamethoxazole / trimethoprim , (78.18%) for cefotaxime , (74.55%) for ceftazidime , (70.91%) for aztreonam, (70.91%) for cefepime,(60%) for ciprofloxacin, (46.3%)  for colistin , (45.45%) for levofloxacin,(42.73%)  for gentamicin,(29.09%)for amikacin and (21.82 %)  for imipenem

2.We defined ESBL, blaCTX-M, blaTEM, and blaSHV as the following Extended-spectrum beta-lactamases (ESBL), blaCTX-M gene, blaTEM gene, blaSHVgene.

Research Manuscript Sections:

Research Manuscript Sections are rearranged according to guidelines outlined in the 'Instructions for Authors' on the journal website as the following: Introduction , Results, Discussion, Materials and Methods, Conclusions.

2. Introduction : Review English language Reference numbers are placed in square brackets at the end of each text.

Results : We edit number of tables and figures, One figure is deleted from line(161) . Edit space and spelling.

Discussion, Review English language Reference numbers are placed in square brackets .

Materials and Methods :

We edit number of tables and figures Review English language Reference numbers are placed in square brackets

 References: The reference list done by the ACS style guide (MDPI.ens references style) , using [End note program] .

Funding, Acknowledgments and Conflicts of Interest are added

Reviewer 3 Report

Antibiotics- Rev.V2: “Multiple drug resistance patterns in various phylogenetic groups of hospital acquired uropathogenic E. coli isolated from cancer patients”

The aim of the study was to describe the prevalence of ESBL-producing E. coli, and their genotypic characteristics, from hospital acquired urinary tract infections from cancer patients in Egypt.

General observations

The article highlights findings on the isolation of ESBL-producing E. coli from hospital acquired urinary tract infections from cancer patients in Egypt. In particular, the prevalence of ESBL-producers among 110 uropathogenic E. coli isolates was 38.18%. All 110 isolates were MDR. CTX-M producers and isolates belonging to phylogenetic group B2 were the most diffused.

However, there are many works describing the same items in the scientific literature, showing high quality of data in particular from both genotypic and phenotypic point of view. Hence, in my opinion, this work have to be improved a lot to consider it for publication.

Major criticisms:

- Basing on both CLSI or EUCAST criteria, the susceptibility profile for colistin cannot be investigated by disc diffusion method. High resistance prevalence (46% of isolates) have to be demonstrated by broth microdilution method (gold standard and the only validated method). If confirmed, this data should be supported by molecular characterization of the resistance mechanism.

- Susceptibility pattern for ciprofloxacin and levofloxacin were very discordant. This could be difficult to explain, since the presence of resistance mechanism to fluoroquinolones make the strain resistant to both antibiotics (especially in E. coli). Hence, these resistance patterns should be investigated more in detail by broth microdilution method.

- Identification algorithm used to identify E. coli isolates should be described more in detail (Figure or table).

- ESBLs should be identified at variant level. Also among the same class of ESBL (CTX-M; TEM; SHV) different variants could have different resistance patterns for cephalosporins.

- Only 42 out of 51 suspected ESBL-producers have been confirmed. Authors should describe resistance patterns of the remaining 9 isolates. Were they resistant to imipenem? If yes, the presence of carbapenemases could give a false negative result of the double disc synergy test.

- I think that one of the most interesting data of the work is that 21.8% of isolates were resistant to imipenem. Authors should determine the presence of carbapenemases or other resistance mechanisms.

- Authors should indicate the year/period of isolation of E. coli strains.

- No correlation of E. coli isolates with pandemic clones (e.g. ST131) have been mentioned.

- Where did you located intermediate resistance patterns? Together with susceptible or to resistant profiles? This data should be specified in Materials and Methods.

- Please, adjust the second line of Table 1: “Ceftriaxone” instead of “Ceftriaxone aztreonam”.

Author Response

# Reviewers Comments:

# Reviewer 3:

Comment 1:

- Basing on both CLSI and EUCAST criteria, the susceptibility profile for colistin cannot be investigated by disc diffusion method. High resistance prevalence (46% of isolates) has to be demonstrated by broth microdilution method (gold standard and the only validated method). If confirmed, this data should be supported by molecular characterization of the resistance mechanism.

Response to Comment 1:

Thank you for this comment….

Response to comment 1:

  • Yes, it is necessary to approve the obtained disc diffusion results of colistin susceptibility by using MIC method as several studies have found disc diffusion to be a defective method to detect susceptibility to colistin[1, 2] so regarding colistin, The MICs with the broth microdilution reference test was done according to the CLSI reference method [3]. European Committee on Antimicrobial Susceptibility Testing (EUCAST) breakpoints were used for interpretation of colistin MIC results (with a susceptible breakpoint of ≤ 2 mg/liter and a resistant breakpoint of  > 2 mg/liter) [4].

  • The resistance rate of colistin obtained with broth microdilution reference test was (32.73%) instead of(46%). We edit the obtained data in table (1) and table (4).
  • We agree with the reviewer that it is useful to support this data by molecular characterization of the resistance mechanism. However, this molecular characterization was not possible to be done due to lack of funding as we said in the manuscript; this research did not receive any specific grant from funding agencies. We will do it in the future. We added a limitation section at the end of the manuscript.

Comment 2:

- Susceptibility pattern for ciprofloxacin and levofloxacin were very discordant. This could be difficult to explain, since the presence of resistance mechanism to fluoroquinolones make the strain resistant to both antibiotics (especially in E. coli). Hence, these resistance patterns should be investigated more in detail by broth microdilution method.

Response to Comment 2:

Thank you for this comment…

  • The susceptibility of isolated strains to ciprofloxacin and levofloxacin was also checked by broth microdilution method as you recommended and concurring results were obtained.

  • The Susceptibility pattern for ciprofloxacin and levofloxacin in our study was similar to some studies such as that reported by Mangal et al., 2017 [5] where the resistance rate for ciprofloxacin and levofloxacin were (82%) and (16%), respectively. Also similar finding was observed in the study reported by Nzalie et al., (2016) [6] where the resistance to ciprofloxacin was 21.4 % and that of levofloxacin was 7.1 %. Drugeon et al., (1999)[7] reported that Levofloxacin is less probable to develop resistant strains compared with older quinolone

  • Drago et al. [8] found that the resistance to the ciprofloxacin was generally more frequent than the resistance to levofloxacin, when the activities of levofloxacin and ciprofloxacin were compared by evaluating their MIC against bacterial uropathogens including coli

Comment 3:

Identification algorithm used to identify E. coli isolates should be described more in detail (Figure or table).

Response to Comment 3:

Thank you for this comment…..

Identification algorithm used to identify E. coli isolates as reported by  Holt et al.,1994[10].

Figure: Identification algorithm used to identify E. coli isolates

Abbreviations: MAC (MacConkey agar) medium,  EMB (Eosin methylene blue) medium

Comment 4:

ESBLs should be identified at variant level. Also among the same class of ESBL (CTX-M; TEM; SHV) different variants could have different resistance patterns for cephalosporins

Response to Comment 4:

Thank you for this comment…

  • In our study, we determined the main classes of ESBLs (CTX-M; TEM; SHV) as earlier reports stated that the most prevalent type of ESBL genes is SHV, TEM, and CTX-M. We agree with the reviewer Although it would be very helpful, studying different variants among the same class of ESBLs was beyond the financial capacity of the researchers. We added a limitation section at the end of the manuscript.

Comment 5:

-Only 42 out of 51 suspected ESBL-producers have been confirmed. Authors should describe resistance patterns of the remaining 9 isolates. Were they resistant to imipenem? If yes, the presence of carbapenemases could give a false negative result of the double disc synergy test.

Response to Comment 5:

Thank you for this comment…..

  • The remaining 9 isolates in our study were sensitive to imipenem.

Comment 6:

- I think that one of the most interesting data of the work is that 21.8% of isolates were resistant to imipenem. Authors should determine the presence of carbapenemases or other resistance mechanisms.

Response to Comment 6:

Thank you for this comment….

  • Due to low resistance level to imipenem (21.8%) compared to the other antibiotics and this didn't pay attention to detect the resistance mechanisms. We gave more concern to cephalosporins since it's the most common antibiotics used in this hospital and our community but it's possible to do this in the future, if the resistance level increases especially that we've recommended this drug as an alternative therapy.

Comment 7

- Authors should indicate the year/period of isolation of E. coli strains.

Response to Comment7:

Thank you for this comment….

  • Urine samples were obtained from cancer patients suffering from urinary tract infection admitted at Assuit university hospitals from March 2018 to March 2019. We added this peroid in the Materials and Methods

Comment 8

- No correlation of E. coli isolates with pandemic clones (e.g. ST131) have been mentioned.

Response to Comment 8

Thank you for this comment….

  • Strains of ST131 coli have two common antibiotic resistance characteristics with the popular showing resistance against fluoroquinolone , while a minority shows resistance to extended spectrum cephalosporins (the main scope of our study) due to extended-spectrum lactamase (ESBL) production [9] so it did not pay our attention but we agree with the reviewer that it is useful to correlate E. coliisolates with pandemic clones (e.g. ST131). This correlation was not possible to be done due to lack of funding as we said in the manuscript; this research did not receive any specific grant from funding agencies. We will do it in the future.

Comment 9

- Where did you located intermediate resistance patterns? Together with susceptible or to resistant profiles? This data should be specified in Materials and Methods.

Response to Comment 9:

Thank you for this comment….

  • Intermediate resistance patterns together with resistant profiles . specification of this data in Materials and Methods was done.

Comment 10:

- Please, adjust the second line of Table 1: “Ceftriaxone” instead of “Ceftriaxone aztreonam”.

Response to Comment 10:

Thank you for this comment….

The adjustment of the second line of Table 1: “Ceftriaxone” instead of “Ceftriaxone aztreonam, was done.

References:

  1. Galani, I., et al., Colistin susceptibility testing by Etest and disk diffusion methods. Int J Antimicrob Agents, 2008. 31(5): p. 434-9.

  1. Lo-Ten-Foe, J.R., et al., Comparative evaluation of the VITEK 2, disk diffusion, etest, broth microdilution, and agar dilution susceptibility testing methods for colistin in clinical isolates, including heteroresistant Enterobacter cloacae and Acinetobacter baumannii strains. Antimicrob Agents Chemother, 2007. 51(10): p. 3726-30.

  1. Wayne, P. Methods for Dilution Antimicrobial Susceptibility Tests for Bacteria That Grow Aerobically. Approved standard 10th edition. Clinical and Laboratory Standards Institute, CLSI document M07-A10; 2015.

.

  1. EUCAST. The European Committee on Antimicrobial Susceptibility Testing. Breakpoint tables for interpretation of MICs and zone diameters. Version 7.1. 2017.

  1. Mangal, N.; Vyas, A.; Kumar, M.; Dalal, A. Antibiogram of Escherichia coli Isolates from Community Acquired Urinary Tract Infection: Special Reference to Fluoroquinolones Resistance. Int. J. Curr. Microbiol. App. Sci 2017, 6, 22-31.

6.Nzalie, R.N.-t.; Gonsu, H.K.; Koulla-Shiro, S. Bacterial etiology and antibiotic resistance profile of community-acquired urinary tract infections in a Cameroonian city. International journal of microbiology 2016, 2016

7.Drugeon, H.B., M.E. Juvin, and A. Bryskier, Relative potential for selection of fluoroquinolone-resistant Streptococcus pneumoniae strains by levofloxacin: comparison with ciprofloxacin, sparfloxacin and ofloxacin. J Antimicrob Chemother, 1999. 43 Suppl C: p. 55-9.

  1. Drago, L.; De Vecchi, E.; Mombelli, B.; Nicola, L.; Valli, M.; Gismondo, M. Activity of levofloxacin and ciprofloxacin against urinary pathogens. Journal of Antimicrobial Chemotherapy 2001, 48, 37-45.

  1. Rogers, B.A., H.E. Sidjabat, and D.L. Paterson, Escherichia coli O25b-ST131: a pandemic, multiresistant, community-associated strain. J Antimicrob Chemother, 2011. 66(1): p. 1-14.

  1. Holt, J.G.; Krieg, N.R.; Sneath, P.H.; Staley, J.T.; Williams, S.T. Bergey’s manual of determinative bacteriology. 9th. Baltimor: William & Wilkins 1994

Round 2

Reviewer 1 Report

The questions are answered well, and I have no more questions.

Author Response

Thanks for your report.

Reviewer 3 Report

Required revisions have been satisfied